# Breeding of Buckwheat to Reduce Bitterness and Rutin Hydrolysis

**DOI:** 10.3390/plants10040791

**Published:** 2021-04-16

**Authors:** Tatsuro Suzuki, Toshikazu Morishita, Takahiro Noda, Koji Ishiguro, Shiori Otsuka, Kenjiro Katsu

**Affiliations:** 1National Agriculture and Food Research Organization, Kyushu Okinawa Agricultural Research Center, Suya 2421, Koshi, Kumamoto 861-1192, Japan; kkatsu9699@affrc.go.jp; 2National Agriculture and Food Research Organization, Institute of Crop Science, Radiation Breeding Division, 2425 Kamimurata, Hitachi-Omiya 319-2293, Japan; tosikazu@affrc.go.jp; 3National Agriculture and Food Research Organization Hokkaido Agricultural Research Center, Memuro Upland Farming Research Station, Shinsei, Memuro, Kasai-Gun, Hokkaido 082-0081, Japan; noda@affrc.go.jp (T.N.); kuro@affrc.go.jp (K.I.); otsukas517@affrc.go.jp (S.O.)

**Keywords:** buckwheat, bitterness, rutin, rutinosidase

## Abstract

Buckwheat (*Fagopyrum esculentum*) is recognized as an important traditional crop in some regions, and its taste is an important characteristic. Of the three cultivated buckwheat species, Tartary buckwheat (*Fagopyrum tataricum*) and perennial buckwheat (*Fagopyrum cymosum*) have strong bitterness in their seeds, which has prevented the wider use of the seeds of these varieties. In Tartary buckwheat, some studies have focused on the cause of strong bitterness generation. Tartary buckwheat seeds contain large amounts of the functional compounds rutin and rutinosidase, and rutin hydrolysis by rutinosidase has been found to be the trigger of rutin hydrolysis. Therefore, a variety with only a trace of rutinosidase and with reduced bitterness is required. The rutinosidase in Tartary buckwheat seeds consists of two major isozymes with very similar enzymatic characteristics, which can hydrolyze flour rutin within several minutes after the addition of water. Recently, the trace-rutinosidase variety Manten-Kirari in Tartary buckwheat was developed. The trace-rutinosidase characteristics were dominated by a single recessive gene. In ‘Manten-Kirari’ dough and foods, such as breads, confectionaries, and noodles, the rutin residual ratio was higher and bitterness was reduced compared to that of the normal-rutinosidase variety. In this review, we summarize the detailed research on the breeding of buckwheat related to reducing bitterness and rutin hydrolysis.

## 1. Introduction

Buckwheat is an important cereal in several countries, such as Japan, China, Russia, and in Europe [1,2]. Buckwheat is recognized as an important traditional crop in some regions [1,2], and its taste is an important characteristic. Generally, sweetness, saltiness, sourness, bitterness, and ‘umami’ are important elements for food taste. For example, the typical stimuli are sugars (sweetness), sodium chloride (saltiness), hydrochloric acid (sourness), and either caffeine, quinine, or some kind of flavonoid (bitterness) [3,4,5,6]. Among them, bitterness is also an important trait for buckwheat. In the *Fagopyrum* genus, three major species are cultivated: common buckwheat (*Fagopyrum esculentum)*, Tartary buckwheat (*F. tataricum*), and perennial buckwheat (*F. cymosum*). The degree of bitterness in seeds varies by species. Common buckwheat, which is cultivated over the largest area in the world [7], has only slight bitterness in seeds, flour, and leaves compared to the other species. On the other hand, Tartary buckwheat has strong bitterness in seeds [8]. Tartary buckwheat has been called ‘bitter buckwheat’, and it has many advantages compared to other species, such as self-pollination and high seed yield in the Hokkaido prefecture in Japan [9]. In addition, Tartary buckwheat is considered a healthy food because it contains a greater amount of rutin compared to common buckwheat. Tartary buckwheat grains have a higher antioxidative activity compared to common buckwheat grains. The contribution of rutin to antioxidative activity was about 90% [10]. Rutin is a flavonol with a high antioxidative effect [11,12]. It is widely found in plants, mainly in leaves and flower buds, although buckwheat is known as the only cereal that contains rutin in its seeds. However, strong bitterness is an important unfavorable characteristic that is widely rejected both by people and animals [5,6]. Therefore, the strong bitterness of Tatary buckwheat seeds has prevented its wide use. Rutin hydrolysis has been identified as a mechanism for the generation of strong bitterness in Tartary buckwheat and it is catalyzed by strong rutinosidase activity in seeds (Figure 1) [13]. Rutinosidase is sufficient to hydrolyze the rutin contained in buckwheat flour within a few minutes after the addition of water [13,14,15]. Recently, a trace-rutinosidase variety was developed in Tartary buckwheat, and showed reduced bitterness [16,17,18,19]. In this review, to improve studies about breeding of buckwheat in terms of preventing bitterness generation and rutin hydrolysis, we summarized the present status and future prospects of buckwheat in terms of the mechanism and breeding for reducing bitterness and rutin hydrolysis.

## 2. Bitterness Generation and Rutin Hydrolysis in Buckwheat

Bitterness is an important quality characteristic, even in buckwheat. In common buckwheat flour, sensory analyses show that bitterness increases with the storage temperature of the flour [20], and this feature has also been observed in instrument evaluation. However, varietal differences in bitterness in common buckwheat have not yet been reported. Several reports have shown that Tartary buckwheat has a strong bitter taste in its seeds and processed foods, such as noodles. Quercetin, the hydrolysis product of rutin by rutinosidase, has been identified as the principle bittering compound in several fruits [21,22]. Kawakami et al., 1995 [23] reported that Tartary buckwheat dough contains at least three bitter compounds: unidentified compounds F3 and F4 and quercetin. Among them, quercetin is the bitter compound generated by rutinosidase activity. Suzuki et al., 2014 [16] hypothesized that rutin hydrolysis triggers bitterness generation. They prepared steam-treated Tartary buckwheat flour in which rutinosidase was completely inactivated, and they also prepared a highly purified rutinosidase from Tartary buckwheat flour. The rutinosidase-free Tartary buckwheat flour and the purified rutinosidase did not have any bitterness. Then, they mixed purified rutinosidase and rutinosidase-free Tartary buckwheat flour and confirmed the generation of strong bitterness. Although it remains unclear which of these three compounds is the major cause of bitterness in Tartary buckwheat, the findings clearly demonstrated that rutin hydrolysis leads to the generation of strong bitterness in Tartary buckwheat flour. The enzymatic kinetics of purified rutinosidase were characterized. Tartary buckwheat rutinosidase consists of two major isozymes, and the molecular weights are 89,000 for both isozymes on gel filtration and 58,200 and 57,400 on SDS-PAGE [15]. The isozymes have generally the same properties. The *K*_m_ values for rutin are lower than those of other flavonol glycosides, indicating that rutinosidase has a high affinity for rutin. In addition, the optimal pH for rutin was 5.0 and the optimal temperature was 40 °C, which are commonly employed as food processing conditions. On the other hand, Yasuda et al., 1994 [14] also purified enzymes which have rutinosidase activity (Rutin degrading enzyme; RDEs). The molecular weights of RDEs were 70,000 on gel filtration and 68,000 on SDS-PAGE and the kinetic constants (RDEs: Km values for rutin 130 and 120 mM) were quite different. Although these differences exist between f3g and RDEs, why do these differences exist? Suzuki et al., 2002 [15] hypothesized that the differences may be responsible for the variations in the cultivated conditions and/or in cultivars. More work is required to clarify this subject. The rutinosidase activity in Tartary buckwheat seed and flour is too strong to inactivate under dry conditions, and it can retain activity after 80 °C for 30 min [24]. Although food-processing technology, such as heat treatment of the seeds or flour, can prevent rutin hydrolysis, it leads to the deterioration of texture [25], color, and flavor and adds costs. Therefore, developing a reduced rutinosidase variety is an important requirement.

To control the bitterness in buckwheat, the rutin content is also an important factor because rutin hydrolysis triggers bitterness generation as described above. To date, many beneficial health effects have been reported for rutin, including the promotion of antioxidative properties [11,26,27], human capillary strengthening [28], antihypertensive properties, anti-inflammatory properties, and alpha-glucosidase inhibitory activities [29]. Therefore, rutin- and rutin-containing foods are considered to be major antioxidants because they show antioxidative activity [12,30]. Many genetic resources have been analyzed, and common buckwheat grains were found to contain 10–30 mg/100 g DW rutin [31,32,33,34,35,36,37]. The possible roles of rutin in buckwheat plants have also been studied, and rutin has a function in the plant defense mechanisms against worm predation, UV radiation, desiccation stress, and cold stress [38,39]. Therefore, the increase in rutin content is also an important breeding objective. The heritability of the seed rutin content was estimated by Kitabayashi et al. [32]; the heritability was as high as that of the days to first flowering. They also performed a parent-offspring correlation analysis for seed rutin content. The correlation coefficients between the parents and progeny lines were almost zero, which indicates that increases in genetic variation, even within a variety through crosses between strains or mutagen treatments, should be effective for the breeding of the seed rutin content by individual selection. To date, common buckwheat varieties with high rutin have been developed. ‘SunRutin’ [40], ‘Toyomusume’ [41], and ‘Ruchiking’ [42] were developed by individual or mass selection. ‘SunRutin’ was developed to increase rutin concentration by selection from ‘Botan-Soba’ by TAKANO CO., LTD. (Headquarters 137 Miyada-mura, Kamiina-gun, Nagano Prefecture 399-4301, Japan). ‘Toyomusume’ was also developed to increase rutin concentration by selection from ‘Kuzuu-Zairai’ by the National Agriculture and Food Research Organization (3-1-1 Kannondai, Tsukuba, Ibaraki 305-8517, Japan). Studies on the rutin synthesis pathway are also important for increasing the rutin content. The pathway of rutin synthesis in buckwheat is as follows: (1) glycosylation by quercetin 3-O glucopyranosyl transferase reaction (3GT) and (2) rhamnosylation by quercetin 3-O glucopyranose rhamnosyl transferase (RT). UDP-rhamnose and UDP-glucose are sugar donors for this reaction. 3GT in buckwheat was purified and characterized using common buckwheat cotyledons [43]. 3GT was estimated to be a 56.0 kDa monomeric enzyme, and the *K*_m_ was 27 μM for quercetin and 1.04 mM for UDP-Glc. In addition, 3GT does not react with other sugar acceptors. On the other hand, the 3GT of other plants generally shows a high affinity, not only for quercetin but also for other flavonols, flavones, and flavanones. Therefore, the common buckwheat 3GT in cotyledon may have evolved specifically for rutin synthesis.

## 3. Development of a Buckwheat Variety with Reduced Bitterness and Rutin Hydrolysis

From the above background, a Tartary buckwheat variety with reduced bitterness was developed [17]. Before developing the variety, the researchers developed a satisfactory screening method to distinguish each rutinosidase isozyme using a rutin–copper complex with native polyacrylamide gel electrophoresis [8]. Using this method, they screened approximately 200 genetic resources and 300 ethyl methanesulfonate mutant lines. Then, they identified four genetic resources of Tartary buckwheat with trace-rutinosidase isozymes from Nepal germplasms. Among the identified individuals with trace-rutinosidase activity in seeds, line ‘f3g-162’ was selected for genetic analysis to identify the loci responsible for this trait. They also performed progeny analysis of the hybrids between ‘f3g-162’ and ‘Hokkai T8’ to clarify the heredity of the trace-rutinosidase trait, while the rutinosidase activity, rutin concentration, and bitterness of the seeds of the obtained hybrids were investigated. In rutinosidase activity, the progeny were clearly divided into two groups. Among the 157 F2 progeny, 115 were normal rutinosidase individuals and 42 were trace-rutinosidase individuals. A segregation pattern corresponding to a 3:1 ratio (normal rutinosidase: trace rutinosidase) suggested that the trace-rutinosidase trait is dominated by a single recessive gene. They also evaluated the bitterness of dough prepared from seeds of both rutinosidase groups. Among all the dough samples, those from normal rutinosidase individuals were uniformly found to have strong bitterness while none of the panelists detected bitterness in the dough samples from the trace-rutinosidase seeds. Although ‘f3g-162’ is a promising breeding line to develop reduced-bitterness varieties, its agronomical characteristics, such as maturing time and yield, are only autumn ecotypes; therefore, it is not suited for cultivation in high latitude regions.

Therefore, the researchers tried to develop agronomical characteristics by artificial crossing between ‘f3g-162’ and ‘Hokkai T8’, which is a leading variety in high latitude regions in Japan. ‘Hokkai T8’ was developed by the National Agriculture and Food Research Organization (3-1-1 Kannondai, Tsukuba, Ibaraki 305-8517, Japan) by single plant selection from the Russian variety ‘Rotundatum’ with high yield, lodging resistance, and suitable for food processing such as roasted seed tea [44]. Tartary buckwheat is a self-pollinating plant that does not show outcrossing. Therefore, they developed a hot water emasculation method in which crossing can be performed efficiently [45]. After crossing, individuals with trace-rutinosidase activity were selected from the F2 progeny, and a promising breed ‘Mekei T27’ was developed by individual selection and line selection. ‘Mekei T27’ was propagated and submitted to variety registration in 2012 under the name ‘Manten-Kirari’ and officially registered as a variety of Tartary buckwheat with the Japanese Ministry of Agriculture, Forestry and Fisheries [46]. The authors evaluated the agronomic characteristics of ‘Manten-Kirari’ by performance tests. In Hokkaido Prefecture, which is in the northern area of Japan, the optimal sowing period for buckwheat is approximately early June. Compared to its parent ‘f3g-162’, ‘Manten-Kirari’ had an earlier maturing time and higher grain yield. In addition, the flour milling percentage and rutin concentration were also higher than those of ‘f3g-162’. In addition, the in vitro rutinosidase activities of ‘Manten-Kirari’ were two orders of magnitude lower than those of varieties with normal rutinosidase activity, including ‘Hokkai T8’. They also investigated and compared rutin hydrolysis during the storage of dough prepared from ‘Manten-Kirari’ with that of ‘Hokkai T8’. In ‘Hokkai T8’ dough, rutin was completely hydrolyzed within 10 min after the addition of water. On the other hand, rutin was only partially hydrolyzed in ‘Manten-Kirari’ dough, even 6 h after the addition of water. They also investigated the bitterness of flour prepared from ‘Manten-Kirari’ flour by conducting sensory evaluations. For the ‘Hokkai T8’ flour, 27 of the 29 panelists detected strong bitterness, while for the ‘Manten-Kirari’ flour, none of the panelists reported bitterness.

To obtain basic information to enable the use of ‘Manten-Kirari’, studies have included time course analyses of rutin hydrolysis in doughs with different blending ratios of Tartary buckwheat and water [18,19]. The residual rutin ratio of ‘Hokkai T8’ (normal rutinosidase variety) decreased rapidly immediately after the addition of water. At 30 min after water addition, the residual rutin ratio was only approximately 10% in samples with a 33% water to flour ratio and 30% blending ratio of Tartary buckwheat flour. In other samples, the residual rutin ratio approached zero. Brunori et al., 2013 [47] also demonstrated similar results. On the other hand, the residual rutin ratio of ‘Manten-Kirari’ decreased slowly compared to that of the normal rutinosidase variety ‘Hokkai T8′. More than 50% of the rutin remained in samples with a water-to-flour ratio <60%. The residual rutin ratio was related to the water to flour ratio, and samples with a 33% water to flour ratio showed a higher residual rutin ratio compared to those with ratios of 60% to 400%, which indicates that a lower water content can delay rutin hydrolysis during food processing. These researchers also evaluated rutin-rich breads produced using this novel variety of buckwheat [19]. The residual rutin ratios in white bread, butter-enriched rolls, pound cake, and galettes were 49.8%, 31.0%, 88.5%, and 26.6%, respectively. Among these foods, pound cake contained the highest residual rutin ratio, while galettes, which have a 400% water to flour ratio, had the lowest residual rutin ratio. The researchers pointed out that differences in the water-to-flour ratio may be associated with differences in the residual rutin ratio between pound cakes and galettes because pound cakes had the lowest water-to-flour ratio while galettes had the highest ratio. The thermal degradation point of rutin is 214 °C [48], and the authors suggested that the difference in cooking conditions (temperature) did not influence the rutin content. Suzuki et al. [18] demonstrated the possibility of making rutin-rich and non-bitter noodles using “Manten-Kirari” flour and investigated the residual rutin ratio in noodles, such as soba noodle and pasta, containing Tartary buckwheat flour. In raw noodles, the blending ratio of Tartary buckwheat flour was 50% for soba noodles and 15% for pastas. In dried noodles, this ratio was 30% for soba noodles and 15% for pastas. The mixed water content was 47% for raw soba noodles and 40% for dried soba noodles. In pasta, a 50% volume of egg was added instead of water. In ‘Hokkai T8’, rutin was hydrolyzed almost completely in all noodles tested. In contrast, approximately 90% of rutin remained in ‘Manten-Kirari’-containing noodles. In addition, while ‘Hokkai T8’ noodles exhibited strong bitterness, ‘Manten-Kirari’ noodles lacked or had only slight bitterness. These results indicate that ‘Manten-Kirari’ holds promise as a material for rutin-rich noodles with minimal bitterness. Moreover, rutin-rich noodles and cookies made with Manten-Kirari have high antioxidative activities. Nishimura et al. [49] investigated whether the rutin-rich Tartary buckwheat cultivar ‘Manten-Kirari’, a trace-rutinosidase variety, could reduce arteriosclerosis, display antioxidant effects, and change body composition in a double-blind placebo-controlled study. Recently, the effect of grain moisture contents on the milling characteristics in ‘Manten-Kirari’ was demonstrated [50], with the authors showing that the rutin content of flour and bran can be controlled by adjusting the grain moisture content before roll milling. Furthermore, this research determined that Tartary buckwheat bran contains extremely high rutin content and thus appears to be a good rutin resource. According to our more recent study [51], roasted Tartary buckwheat ‘Manten-Kirari’ bran with a high rutin content, which is promising for the development of rutin-rich foods, was successfully obtained. Tartary buckwheat bran is an underutilized byproduct of buckwheat flour production that is rich in rutin. To obtain rutin-rich Tartary buckwheat bran, the relationship between rutin content and color change in Tartary buckwheat bran samples of amounts of 20 g to 100 g during roasting between 10 min to 30 min at 160 °C to 240 °C was investigated. Increasing the roasting time and temperature and decreasing the sample amounts caused reductions in the L* and b* values and the rutin content. However, even after roasting for 10 min at 160 °C to 230 °C and for 20 min at 160 °C in sample amounts of 100 g, Tartary buckwheat bran retained high levels of rutin. As a higher rutin content has been closely associated with higher values of L* and b*, analyzing color parameters using a chromaticity meter is promising for predicting the rutin content in roasted Tartary buckwheat bran. The investigation of the safety of varieties is also important. The developers also studied the acute, subacute, and mutagenicity potential of the variety using experimental animals and bacteria [52,53], and did not observe any unsafe varieties. These reports indicate that the trace-rutinosidase Tartary buckwheat variety has many advantages compared to the normal rutinosidase variety. In Japan, more than 80% of the Tartary buckwheat cultivation area changed to ‘Manten-Kirari’ within five years after the release of the variety.

The trace-rutinosidase activity in ‘Manten-Kirari’ in dough gradually hydrolyzes rutin. In addition, contamination of Manten-Kirari seeds or flour from varieties with normal rutinosidase activity sometimes occurs, such as during sowing, harvesting, and food processing. Suzuki et al. [54]. investigated the effects of different variables (such as water content, dough storage temperature, and blending ratio of Tartary buckwheat flour) on the rutin residual ratio. In addition, they also tested the effects of the addition of sodium bicarbonate (NaHCO_3_) on rutinosidase activity in buckwheat dough. As a result, the addition of 0.5% NaHCO_3_ can prevent rutin hydrolysis almost completely in dough made from ‘Manten-Kirari’. When contamination rates from normal rutinosidase varieties were less than 1%, the addition of NaHCO_3_ to dough made from Manten-Kirari under low-temperature storage of dough (5 °C) retained a rutin residual ratio over 80% with a water content of 30% for up to 2 h after the addition of water. On the other hand, rutin hydrolysis cannot be prevented in dough containing high water content, such as in galletes. For future studies, a variety without rutinosidase is required. Of the three cultivated species, only common buckwheat does not show rutinosidase activity. Therefore, interspecific hybridization between common and Tartary buckwheat or common and *F. cymosum* is important.

## Figures and Tables

**Figure 1 plants-10-00791-f001:**
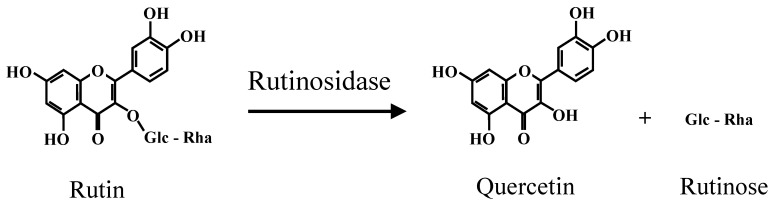
Rutin hydrolysis in Tartary buckwheat flour.

## Data Availability

The data that support the findings of this study are available on request from the corresponding author. The data are not publicly available due to privacy or ethical restrictions.

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
