# Peer review of "Breeding of Buckwheat to Reduce Bitterness and Rutin Hydrolysis"

_plants, 2021, doi:10.3390/plants10040791_

Round 1

Reviewer 1 Report

The article is clear. The subject matter is interesting. However, for a review article, it is too short. I would suggest expanding the article. Please, include illustrations. Latin names should be included in the abstract and not just common names.   According to author Contributions, only one author T. Suzuki designed and wrote the manuscript. I find it incomprehensible that as many as five additional people are co-authors just for critically correcting the text. I would have expected more input from these additional authors.

Author Response

Thank you very much for the thoughtful and constructive feedback you provided regarding our manuscript. We also appreciate the time and effort you and each of the reviewers have dedicated to providing insightful feedback on ways to strengthen our paper. Thus, it is with great pleasure that we resubmit our article for further consideration. We agree with you and revised our paper according to your suggestion.

(Please refer attached file)

Reviewer 2 Report

I had difficulties in understanding how this can be an article. It is probably a review as judged by some sentences in the abstract, but this needs to be clearly stated. The abstract mentions a “summarize detailed research on the breeding of buckwheat related to reducing bitterness and rutin hydrolysis”. However, this is hardly comprehensive for the readers.

Some major concerns:

  • What are the aims of this review? What are the hypothesis or questions that triggered this study?
  • The introduction is full of explanatory sentences that need to be backup with references; otherwise, they seem personal statements:

Line 38: Tartary buckwheat has been called ‘bitter buckwheat’, and it has many advantages compared to other species, such as self-pollination and high seed yield.

Line 41: Tartary buckwheat is considered a healthy food because it contains approximately 100 times more rutin than common buckwheat. Tartary buckwheat grains had 3-4 times higher antioxidative activity than common buckwheat grains.

I do understand that explaining the methodology of the cited papers in a review might not be easy, but the authors are mixing results from very different analysis (and methodologies), which should be better explained. 

I could not follow any of the breeding lines mentioned by the authors.

Author Response

(The authors gave the same response as above.)

Round 2

Reviewer 1 Report

I have read the corrected version of this manuscript and it reads much better, authors improved it.

Reviewer 2 Report

Thank you for time in answering to my comments. I have read the second version of this article and it reads much better. Congratulations for this review.